# Circuit Football Training Customized for Young Players during and after the COVID Period

**Dan Gheorghe Păun** [1,*] **, Gheorghe Grigore** [2] **and Laurian Ioan Păun** [3]

1   Faculty of Physical Education and Sport, Spiru Haret University, 041905 Bucharest, Romania
2   Faculty of Physical Education and Sport, National University of Physical Education and Sport, 060056 Bucharest, Romania
3   Lower Secondary School No. 1, 500423 Brasov, Romania
*   Correspondence: danpaun76@yahoo.com or ushefs_paun.dan@spiruharet.ro

**Abstract:** Football was, is, and (perhaps) will always be the most widespread sport all over the world. High performance means hard work for all players, regardless of age. In this respect, training methods are adjusted to the needs of modern football, converging towards the successful formation of players with good technical tactical, physical, theoretical, and psychological levels. In the context of the research, we chose to use a program based on the circuit method, which would lead to the re-adaptation of junior football players to performance demands after the COVID-19 pandemic period (which was very difficult for sports in general). The circuit method is a suitable factor for the current trends in modern football. Our research showed that the circuit method was consistent with the specific goals of football training for 10–12 year olds. The method proposed for this study was not new, but it was applied for the first time to children and juniors, considering their age characteristics and the requirements of the modern world. Its implementation involved using a literature review, active observation, and practical methods. The exercises used in the study were designed in compliance with the specifics of both the age of 10–12 years and the game of football played at this level. The paper demonstrates that the circuit method enables teachers and coaches to achieve the football training goals intended for this age.

**Keywords:** physical fitness; young players; football protocol; circuit method; qualification-related assessment

## 1. Introduction

The spread of football in our society is a huge phenomenon: the game is played on all continents at a constantly increasing level. We do not need reminder of the reason why football is the sport with the largest number of supporters; however, we can say that it is no longer a simple sport but has become a successful business that can generate substantial economic gains. More than ever, a real show attracts the public in increasingly full stadiums or as a TV audience, with over 1 billion viewers of the World Final Championship in Brazil. However, this is only the visible part because what is not seen represents 90% of the whole phenomenon. We state this based on the titanic amount of work behind the football training process (children, players, coaches, and teachers). It should be recalled that, for a football show, the training process starts early, at the age of 6 (and even earlier, according to some sources), following a rather sinuous path [1].

The circuit method proposed for this study is not new, but it is applied for the first time to children and juniors, considering their age characteristics and the requirements of the modern world. Due to the practical means and structures used, this method is considered a suitable factor for the current trends in modern football [2–4]. Our research is an attempt to bring to the fore a possible approach to the training process of junior football players against the background of an unknown global reality (the emergence of a pandemic situation) with negative effects on sports performance.

In this context, the research question is as follows: Can we achieve this through a circuit method with technicaltactical content to improve the degree of efficiency in a competitive game between the autumn and spring stages? We believe that this is possible.

The article is structured in seven sections that address representative categories for the chosen topic as follows: theoretical details based on information obtained from the literature, materials and methods used in the research, presentation of relevant data and descriptive statistics, discussion, conclusion, and limitations of the study.

## 2. Theoretical Background

An analysis shows the effect of the challenges that football players must face on the field that also affect them at a psychological level. Off-field challenges also occur at this, as well as at psychosocial and academic or work-related levels. Therefore, the relevance of successfully coping with these challenges and its importance for proper adaptation to elite football are indisputable. [5]

Challenges are numerous for football players at the levels of children and juniors and are related to sports, school, and social life. Psychological problems can arise as a result of playing football starting at the age of 10–12 years. Age characteristics influence the educational process of forming a child within the football club to which they belong [2–4]. The consequences of these typological features are morphological, functional, and psychological.

High performance requires hard work, organization, discipline, a positive attitude, and a spirit of sacrifice; despite this, not all participants manage to play in the great stadiums of the world. In today's football, the training of children and juniors has become an activity based on in-depth data from biochemistry, biomechanics, medicine, psychology, and physiology [6,7].

Against this background of unique situations, the problem of regaining the form of the sport has emerged. What happens in a new, totally unexpected context, such as a pandemic? The players, either top-level seniors or children and juniors as future performers, had to face totally atypical realities for sports in general and football in particular. Maintaining social distancing, avoiding contact with partners and opponents, training in isolated conditions (through individual and individualized training, where possible, but how many players had such opportunities?), not having spectators at games—how many challenges!

Fortunately, the difficult period passed, but problems appeared shortly after and were related to exercise parameters, sport form, playing style, technical–tactical execution, and re-adaptation to the performance environment.

In order to solve the problems that appeared in post-pandemic training processes with direct implications for the competitive game, we tried to use exercises with technical–tactical content based on the circuit method, which were mainly aimed at the offensive phase and, to a lesser extent, the defensive phase. Considering the reality in which athletes returned to training after the period of forced interruption, two important objectives in designing these exercises based on the circuit method were creativity and attractiveness [4].

The usefulness of the proposed method was verified by applying some game indicators to the investigated group during the 2020–2021 official championship and by analyzing some statistical results (three variables) related to these indicators.

The 30 games (divided into two periods) where data were collected highlighted the need to model technical and tactical training through the circuit method, with positive implications for the players' careers.

The mental training of players aged 10–12 focuses on technical–tactical exercises carried out at a fast pace. As previously mentioned, the psychological problems related to playing football start at this very age. Mentally, the focus is on speed and skill exercises instead of long and boring ones that require more voluntary attention than a child can provide. Physical skills and dynamic stereotypes are repeated continuously, focusing on correctness from the beginning in order to perfect and master them. The volume and intensity of the exercises are adjusted to give them a predominantly playful touch and avoid attracting

attention—concentration and stability—in a tiring manner. It is recommended that a work environment follow the same direction as a game, thus creating positive interpersonal relationships and communication within such a group. Playing football provides children with the opportunity of acquiring a large number of physical skills, but the stages of skill formation must be observed: knowledge of the physical structure, analytical learning, organization and systematization, synthesis and automation, and improvement. During the process of training such skills, the laws of transfer and interference should also be taken into account. Attention during a football game is voluntary, meaning that a child is consciously focused [8,9].

Outdoor classes are essential for the physical, motor, and social development of young football players. Psychomotor and motor skills based on movement games can be developed through the methods used in leisure activities [10]. Thus, we can state that exercise is a means of preventing or reducing many physical or emotional problems. The physical fitness of any young football player is part of their lifestyle and provides good prerequisites for performance [11].

The specific content of football training for children in the studied age range consists of focusing on playful actions based on structures that take the form of competitions played on small-sided fields with simplified rules. As for the selection of exercises and games, the decision is in favor of those with accurate, clear, easy-to-evaluate rules, but mostly those promoting sociability because the "aggressor" seeks to establish contact with others. In this context, the coach will adopts a firm, equidistant position and controls good organization of the training throughout the entire process [12–14].

Therefore, the methodology of training athletes, including young football players, is a long, difficult, and complex process that involves several areas, among which the psychological problems of training and the psychological characteristics of children aged 10–12 years. This requires rigorous planning based on the multilateral training of a coach who aspires to the stated objectives, which must be scheduled, formulated, and controlled [15–17].

However, when completely unusual situations arise (even at a global level), what remains of the rigorous planning, the multilevel training of coaches, and the scheduled goals? In the short term, very little, and in the long term, almost nothing (perhaps only the automatisms created in the technical plane).

In March 2020, the UK's elite football academies closed as a result of the lockdown imposed by the government to keep the COVID-19 pandemic under control. This situation forced parents, players, and coaches to reconsider how they interacted and supported each other. The goals were: (a) to explore the perceptions of players, parents, and coaches (i.e., the athletic triangle) of how they interacted and collaborated during the COVID-19 pandemic to support wellbeing and performance and (b) to identify opportunities to improve the functioning of those included in the athletic triangle through adaptations made following the lockdown [18].

Football players need to use resources and strategies to cope with changes that occur in the field of sports, but also in other areas of life, such as psychological, social, and school ones, thus facilitating successful adaptation to elite sports [19].

In the post-pandemic period, the challenge of identifying opportunities to improve specialized training (which was greatly affected by the lockdown) led us to use the circuit method.

Studies have shown that FIFA 11+ (a complete warm-up program to reduce injuries among football players) can improve fitness components through resistance and neuro-muscular exercises. Currently, resistance training using intensity circuit training (HICT) is considered beneficial for increasing the physical fitness component, including cardiopulmonary fitness [20]. HICT is performed by repetitive weight exercise in a continuous manner, moving from one station to another with a minimal rest interval between stations. It can be a highly effective means of increasing an individual's VO2max, a valid marker of cardiopulmonary health [21]. Stations and exercises are established according

to specific criteria, including the alternate use of large muscle groups and individualized dosing [22–24].

In the current situation, we tried to readjust to the technical–tactical requirements in a physical regime and obtain a sport form that would allow the proposed objectives to be achieved according to an analytical global integrative model.

An important aspect of our specific field of activity was related to the issue of training density. It should be emphasized that density is an indicator of quality in the training of a football player. Therefore, specialized methodology constantly aims to optimize the relationship between different types of density specific to the training process, namely pedagogical density and motor density, which are crucial for highlighting the quality of training [25].

For theoretical aspects, a coach can use exergaming (digital games), which nowadays has become an innovative tool of knowledge transfer and motivation for the young generation [26,27]. Having fun and playing is the most important incentive for young people. In this sense, a coach can use another innovative technology that is loved by children, namely virtual reality (VR). VR is used in football to design training scenes, collect movement data, simulate different actions, evaluate training performance, and monitor physiological, biochemical, and psychological data [28,29].

In a study, differences in visual search and locomotion behavior were examined in a group of skilled football players aged 10–12 years. The participants watched video clips of a 4-to-4 position game, which were displayed on a large screen. They were asked to participate in the game by choosing the best position to receive the ball passed by one of the players in the video. It was concluded that differences in visual search and locomotion behavior could be used as indicators for the identification of talented junior football players [30].

## 3. Materials and Methods

### 3.1. Preliminary Phase of the Research

This study presents a training circuit for young football players aged 11–12 years. Our objective was to find out whether the proposed circuit training method had a positive impact on football players in technical and tactical terms.

The research sample consisted of 16 young football players from the Progresul Bucharest Club, all born in 2009. To carry out this study, we asked and received the consent of each player's parents. We also obtained the approval of the Progresul Bucharest Ethics Committee to conduct training using the circuit method. During the workouts, the coach used ameliorative training methods based on six types of circuits that were quantified according to the number of tackles, takeovers, fouls, interceptions (for the defensive phase), dribbles, assists, and shots (for the offensive phase) performed during competitions [31,32].

Circuit 1. The player controls a ball over a distance of 10 m after a self-pass followed by a forward roll on a gym mat (Figure 1). The next step is to regain possession of the ball by controlling it over a distance of 5 m, which is immediately followed by another self-pass and a bench exercise followed by a new possession of the ball. During the third step, the player stops the ball with the sole of the foot, jumps over three low hurdles (15 cm), and returns to take the ball through four cones. Afterward, the player kicks the ball into a goal.

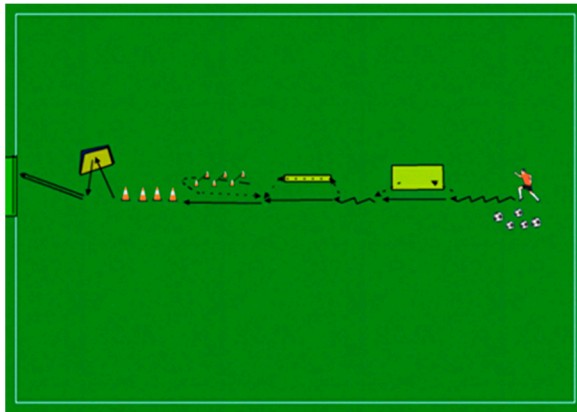

**Figure 1.** Circuit 1.

Circuit 2. The player controls a ball over a distance of 10 m and then kicks it into a small goal. The next step involves a 180-degree turn jump, followed by going through asymmetrically arranged hoops and immediately rolling before shooting the ball into a small goal. Then, the player performs another 180-degree turn jump, passes through a ladder, and shoots again into a small goal (Figure 2).

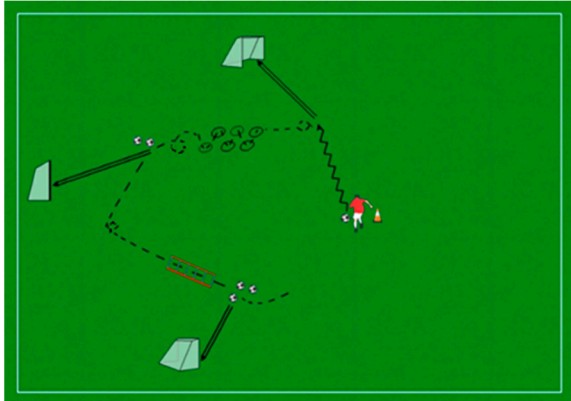

**Figure 2.** Circuit 2.

Circuit 3. The player controls a ball over a distance of 5 m and then shoots at a goal. The player clears a hurdle (×6) from the side, alternately jumps over or passes under three hurdles (50 cm), followed by shooting. Next, the player passes through five poles and shoots (Figure 3).

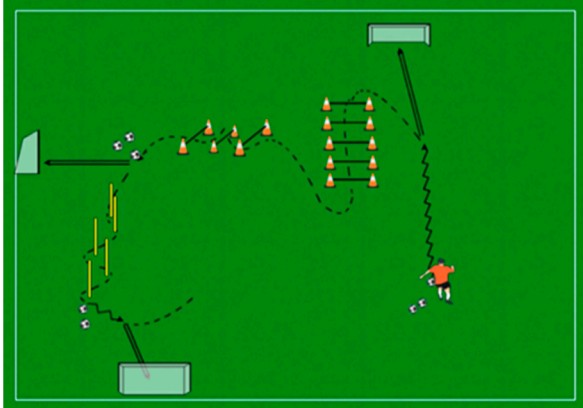

**Figure 3.** Circuit 3.

Circuit 4. The player starts at a signal by rolling forward, passes through asymmetrically arranged hoops, and drives the ball through four cones. Next, the player alternately jumps over or passes under three hurdles (50 cm), which is immediately followed by a forward roll while controlling the ball over a distance of 5 m and then shooting (Figure 4).

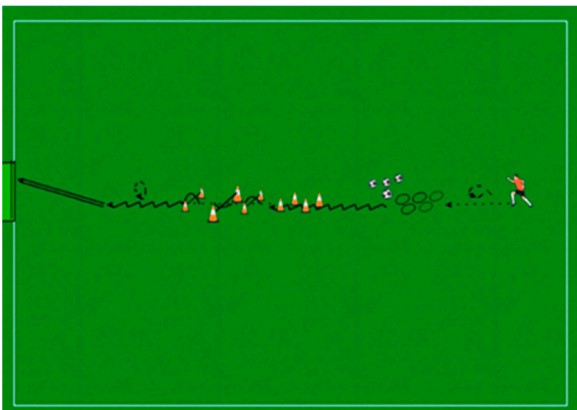

**Figure 4.** Circuit 4.

Circuit 5. The player makes a forward roll followed by a bench jump performed at speed. Next, the player passes through five cones, then through asymmetrically arranged hoops and a ladder, controls the ball over a distance of 5 m, and shoots into a small goal (Figure 5).

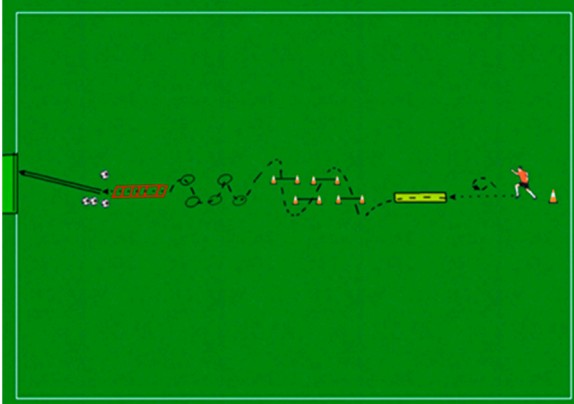

**Figure 5.** Circuit 5.

Circuit 6. Work is simultaneously performed by players. Station 1. The player sprints over a distance of 5 m, skips for 20 s, and then drives the ball and shoots into a small goal. Station 2. The player runs 5 m, passes through randomly arranged cones, drives the ball, and shoots. Station 3. Same as Station 2, but the player simultaneously jumps with both legs into asymmetrically arranged hoops, and then drives the ball and shoots. Station 4. Same as Station 3, but the player performs three alternately forward and backward rolls on a gym mat, drives the ball, and shoots (Figure 6).

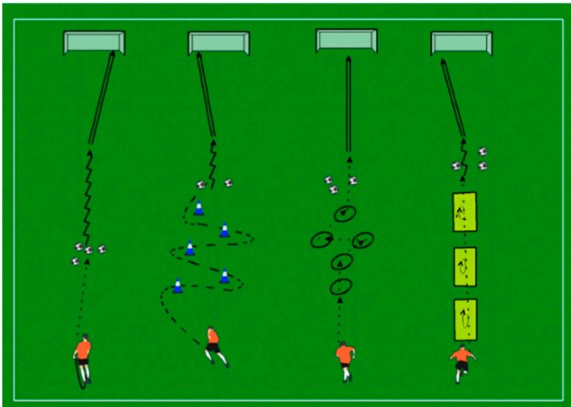

**Figure 6.** Circuit 6.

The purpose of the article was to find out if this type of training had an influence on the overall performance measured by shots on a target and shots on the goal.

The main research questions were as follows:

- Are there differences between matches played in the first and second halves of the championship?
- Are there differences between the defensive and offensive phases in terms of performance?
- Are the training methods appropriate?
- What are the team's weaknesses and strengths?
- What can be improved?

To test the statistical hypotheses, we decided to perform a confirmatory factor analysis (CFA) using structural equation modeling (SEM), which estimated saturation and calculated a series of match indices that described how well the research data fit the model. SmartPls software provides numerous tests that can be used to ensure factor analysis and the coherent interpretation of data to assume the research results. For example, the consistency of our model was based on validation statistics such as composite reliability, Cronbach's Alpha, rho_A, and AVE (average variance extracted).

### 3.2. Design and Research Phases

Considering that this was preliminary research, we only worked with 16 football players. The sample was not representative of the entire country for this age group but was representative of the Progresul Bucharest Club. If the results confirm our hypothesis, we plan to conduct a new study on a larger sample that can be representative of the entire country.

To answer the research questions, continuous observations and data collection were performed by the coach (teacher) during both training and games or competitions.

The training sessions carried out in the first half of the season had technical and tactical content based on simple exercises, finishing with a shot on the goal (examples: a. takeover, driving the ball 20 m, and shooting on the goal; b. passes between two players over the distance of 25 m and shot on the goal). The objective of this stage was to improve a player's ability to finish with a shot on the goal. In the second half of the season, the circuit method was implemented, which increased the complexity, dosing, and difficulty of the training sessions. The objective of this stage was also to improve a player's ability to finish with a shot on the goal, but this time the focus was on speed, collaboration between players, and orientation on the playing field.

Thus, data about some indicators (Table 1) were collected from 15 matches played in the first half of the junior championship in which the team participated, and then six training circuits with technicaltactical content were scheduled and performed over 4 months, during which data were gathered from the matches played in the second half of the championship to be analyzed.

**Table 1.** Name, code, and significance of variables.

| Variable | Sub-Items | Significance |
|---|---|---|
| First half of the season (F) | Min F<br>Tackles F<br>Takeovers F<br>Fouls F<br>Interceptions F<br>Dribbles F<br>Assists F | minutes played in the first half of the season<br>number of tackles<br>number of takeovers<br>regulation infractions<br>number of interceptions<br>number of dribbles<br>number of assists |
| Second half of the season (S) | Min S<br>Tackles S<br>Takeovers S<br>Fouls S<br>Interceptions S<br>Dribbles S<br>Assists S | minutes played in the second half of the season<br>number of tackles<br>number of takeovers<br>regulation infractions<br>number of interceptions<br>number of dribbles<br>number of assists |
| Shots | Goals GF<br>Shots SF<br>Goals GS<br>Shots SS | number of goals scored in the first half of the season<br>number of shots on the goal in the first half of the season<br>number of goals scored in the second half of the season<br>number of shots on the goal in the second half of the season |

The six circuit training sessions were scheduled during the first half of the season as follows:

- February: Day: Thursday—Circuits 1 and 2—30 min (15 + 15);
- March: Day: Thursday—Circuits 1 and 2—20 min (10 + 10); Circuits 3 and 4—30 min (15 + 15);
- April: Day: Thursday—Circuits 3 and 4—20 min (10 + 10); Circuits 5 and 6—30 min (15 + 15);
- May: Day: Tuesday—Circuits 1, 2, and 3—30 min (10 + 10 + 10); Circuits 4, 5, and 6—30 min (10 + 10 + 10).

In this article, we decided to present the collected data regarding the defensive phase (tackles, takeovers, fouls, and interceptions) and offensive phase (dribbles, assists, and shots—scored goals and shots on the goal) in both halves of the championship. In this regard, three main variables were created as follows: the matches played in the first half and second half of the season as the formative variables with seven sub-items and the shots as the reflective variable with four sub-items (Table 1).

The research hypotheses were:

- H1: The indicators with specific content assessed during the first vs. the second half of the season had obvious improvement rates due to the implementation of circuit training with an emphasis on finishing with a shot on the goal.
- H2: Competitive performance, which was highlighted by indicators specific to finishing with a shot on the goal (number of shots on the goal, into the goal area, and scored goals), could be influenced by the effective programming of circuit training (with technical and tactical content).

The representative variables of the research were matches played in the first half of the season, matches played in the second half of the season, and shots (scored goals and shots on the goal). The indicators composing the three above-mentioned variables were intended to reveal to what extent the investigated athletes made progress during competitions after implementing the circuit method (with technicaltactical content) in their training.

## 4. Results

*Descriptive Statistics*

According to our data, the average number of minutes played in the first half of the championship (autumn) was higher than in the second half (spring) (731 vs. 704). The average number of tackles was 94.6 in the autumn and 117.7 in the spring. As the primary purposes of tackling are to dispossess an opponent of the ball, to stop a player from gaining ground towards the goal, or to stop them from carrying out what they intend, we can state that our players improved their technique in the second half of the season. The average number of takeovers increased from 29.9 in the autumn to 33.9 in the spring, which also emphasized an improvement in the players' technique. The average number of faults decreased from 31.4 in the autumn to 26.3 in the spring, which highlighted improvement in the fair-play behavior of athletes. The average number of dribbles was 27.9 in the first half of the season and almost doubled to 41.3 in the second half, meaning that the players had much better control of the ball in direct relation with their opponents. The average number of assists increased from 2.5 in the autumn to 2.8 in the spring, which meant that the players' passes were more accurate and much better controlled and that their tactical levels were also better. The main performance of the football players was assessed by the number of goals scored in the first half of the championship (Goals GF) and the number of shots on the goal in the same first half of the championship (Shots SF). For the former, the average increased from 6.9 in the autumn to 7.9 in the spring, while for the latter, the average decreased from 2.3 in the autumn to 2.1 in the spring, meaning that more shots were associated with goal scoring in the second half of the season, which revealed a more accurate game and control of the ball (Table 2).

The standard deviation and standard error were too high for some indicators, such as Min, Tackles, Assists, Goals, and Shots, but these values were normal and explained by the positions of players (striker, midfielder, defender, and goalkeeper). It is very normal for a goalkeeper not to score because their role is to protect the goal, while a striker usually scores many goals. For the same reason, a goalkeeper has very few tackles, faults, etc.

The above table shows that positive results were obtained in the second half of the season compared to those achieved in the first half, even if the average number of minutes played was higher in the autumn. A possible explanation would be related to situations in which the opponent fragmented the game of the Progresul Bucharest team by finishing more quickly due to better-developed technicaltactical actions (especially after the successful performance of tackles). The concern to attack in order to score more goals was also obvious from both the lower number of fouls and the higher average number of dribbles in direct relation to the opponent. The final result of the training in the presented context was an increase in the average number of scored goals from 6.9 to 7.9, which we considered to be a very good value.

To demonstrate a univariate, normal distribution, values for skewness and kurtosis between $-2$ and $+2$ were regarded as acceptable [33]. According to Hair et al. (2010), data are considered normal if skewness is in the range of $(-2, +2)$, and kurtosis is in the range of $(-7, +7)$ [34]. In our case, the Tackles H, Tackles A, and Fouls A variables did not show normal Gaussian distribution, but this could be explained by the positions of players, as stated before. This is why we decided to apply a PLS-SEM model, which is more suitable for variables with abnormal distribution and nonparametric variables, as well as in the case of a small amount of data (Table 2).

**Table 2.** Descriptive statistics.

| Statistics | Min F | Tackles F | Takeovers F | Fouls F | Interceptions F | Dribbles F | Assists F | Goals GF | Shots SF | Position |
|---|---|---|---|---|---|---|---|---|---|---|
| Mean | 731.1 | 94.6 | 29.9 | 31.4 | 3.1 | 27.9 | 2.5 | 6.9 | 2.3 | 1.8 |
| Standard Error | 89.2 | 24.8 | 3.1 | 2.9 | 0.4 | 3.7 | 0.5 | 1.7 | 0.5 | 0.2 |
| Standard Deviation | 356.9 | 99.2 | 12.2 | 11.5 | 1.7 | 14.8 | 2.0 | 7.0 | 1.8 | 0.8 |
| Kurtosis | −1.4 | 12.4 | −1.7 | −1.5 | −0.2 | −1.2 | −1.9 | −1.8 | −2.1 | −1.1 |
| Skewness | −0.7 | 3.4 | 0.1 | −0.1 | −1.3 | −0.2 | −0.6 | 0.5 | −0.1 | 0.5 |
| Minimum | 140 | 18 | 13 | 13 | 0 | 2 | 0 | 0 | 0 | 1 |
| Maximum | 1050 | 448 | 44 | 44 | 4 | 44 | 4 | 17 | 4 | 3 |
| 95% Confidence Interval | 190.2 | 52.8 | 6.5 | 6.1 | 0.9 | 7.9 | 1.1 | 3.7 | 1.0 | 0.4 |
| | Min S | Tackles S | Takeovers S | Fouls S | Interceptions S | Dribbles S | Assists S | Goals GS | Shots SS | |
| Mean | 704.4 | 117.7 | 33.9 | 26.3 | 2.9 | 41.3 | 2.8 | 7.9 | 2.1 | |
| Standard Error | 92.7 | 51.1 | 5.8 | 4.9 | 0.5 | 7.2 | 0.6 | 1.9 | 0.7 | |
| Standard Deviation | 370.8 | 204.4 | 23.3 | 19.7 | 2.0 | 28.7 | 2.4 | 7.7 | 2.7 | |
| Kurtosis | −1.2 | 15.2 | 2.6 | 9.5 | −0.9 | −1.0 | −1.8 | −0.3 | −0.8 | |
| Skewness | −0.8 | 3.9 | 1.6 | 2.8 | −0.4 | 0.7 | 0.0 | 1.0 | 0.9 | |
| Minimum | 70 | 14 | 8 | 9 | 0 | 10 | 0 | 1 | 0 | |
| Maximum | 1050 | 876 | 97 | 93 | 6 | 94 | 6 | 25 | 7 | |
| 95% Confidence Interval | 197.6 | 108.9 | 12.4 | 10.5 | 1.0 | 15.3 | 1.3 | 4.1 | 1.4 | |

This more reliable regression technique (a) reduces the variance in the endogenous construct residuals; (b) poses minor identification problems; (c) shows effective results even with small sample sizes; and (d) primarily combines formative and reflective constructs [35]. When a structural model is extremely complicated, sample size is limited, CB-SEM (covariance-based structural equation modeling) assumptions are not met, and a model incorporates both formative and reflective constructs, PLS-SEM (partial least squares structural equation modeling) or a path analysis should be used [36]. When a study is focused on prediction or theory development, PLS-SEM is the ideal approach (with contributions to theory development). It is mostly used for predictive analyses and to explain complicated relationships [37].

Considering these two hypotheses, the research used SmartPls [38] to assess consistency through composite reliability, as shown in Table 2. The authorized threshold values for a consistent model were: composite reliability (>0.6), Cronbach's alpha, rho_A (>0.7, as the authorized bottom value), and AVE (>0.5). The Cronbach's alpha coefficients indicated that the questionnaire items were appropriate for our analysis, meaning that the sub-items of the Shots variable were relevant to the model and had a large positive influence. It can be observed that the Shots variable had very high values for all the tests: composite reliability (CR = 0.891 > 0.7), Cronbach's alpha (CA = 0.886 > 0.7), average variance extracted (AVE = 0.805 > 0.5), and rho_A (0.901 > 0.5). The R-squared value (the coefficient of determination) was 0.789, which was higher than the accepted minimum of 0.5. We can say that 79.1% of the variance in the variable was explained by the model. The path coefficient for the Shots variable in the first half of the season was very small (0.253), meaning that this phase had a very small influence on the model fit and overall results. In the second half of the season, the path coefficient for the Shots variable was very high (0.665). Thus, we can state that our football players had poor results (very few accurate shots) in the first half of the championship, but their results improved in the second half after applying the circuit training (Table 3 and Figure 7).

**Table 3.** Validation steps.

| Formative and Reflective Constructs | Composite Reliability | Cronbach's Alpha | AVE | rho_A | R-Squared | Path Coefficients | |
|---|---|---|---|---|---|---|---|
| | (>0.7) | (>0.7) | (>0.5) | (>0.5) | (>0.5) | First Half -> Shots | 0.253 |
| First half of the season | - | - | - | 1 | 1 | | |
| Second half of the season | - | - | - | 1 | 1 | | |
| Shots | 0.891 | 0.886 | 0.805 | 0.901 | 0.752 | Second half -> Shots | 0.665 |

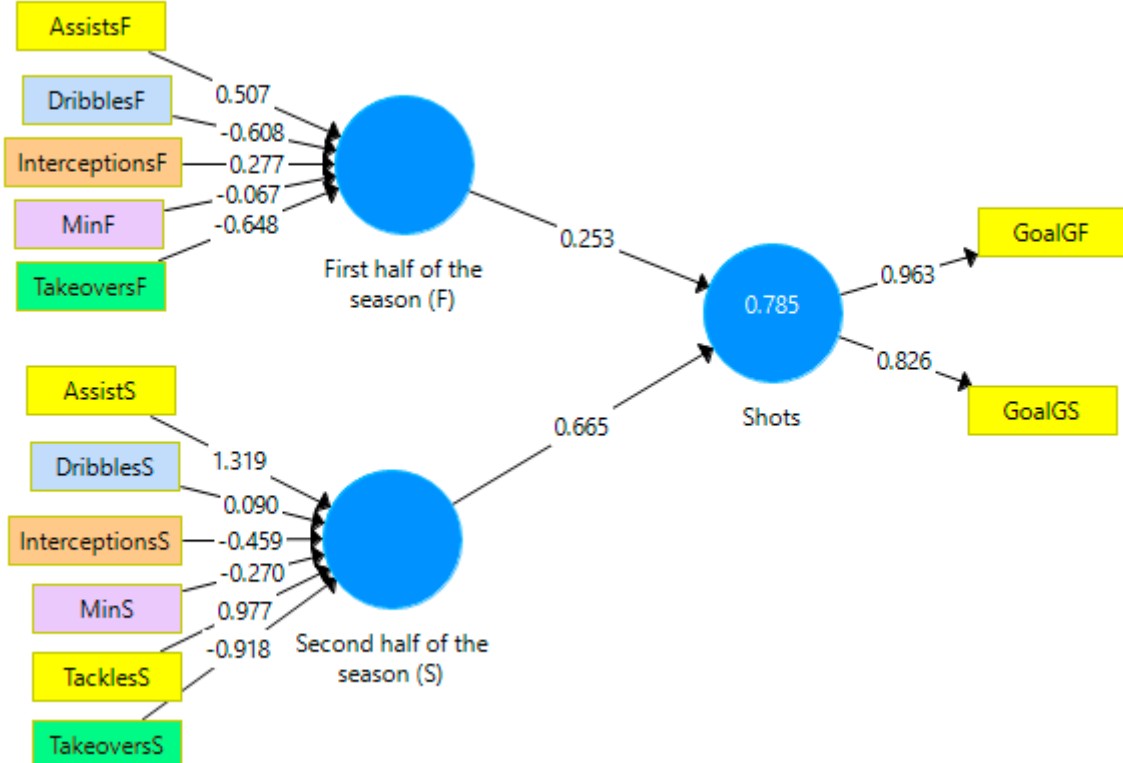

**Figure 7.** Cronbach's alpha analysis and path coefficients.

Figure 7 shows that the loading factors for most of the Shots variables had values greater than 0.7, which proved their impact on the performance weight. It can also be noted that the loading factor (LF) for the Shots variable was 0.826 in the first half of the season and 0.963 in the second half, meaning that the number of shots into the goal area was smaller in the autumn compared to spring. Shooting accuracy improved very much from one half of the championship to the other. There were many shots on the goal in the autumn because children did not play the ball well (they did not perform enough dribbles, tackles, etc.) but were only focused on goal scoring [33].

The main elements that influenced the first half of the season where Takeovers H (LF = −0.648), Dribbles H (LF = −0.608), and Assists H (LF = −0.507), while in the second half, the variables with higher loading factors and greater influence were Assists A (LF = −1.319), Tackles A (LF = 0.977), Takeovers A (LF = −0.918), and Interceptions A (LF = −0.459), as can be seen in Figure 7.

Assists had a significant impact on highlighting the positive results obtained; they showed a more than doubled value in the spring (1.319 vs. 0.507), which indicates that the team managed to finish many more times, thus increasing their chances of shooting on the goal and scoring, which was consistent with the research objectives. The increased number of takeovers in the second half of the season also emphasizes that the means used in the circuit method led to a more elaborate, more tactical game for the aim pursued: goal

scoring and victory. However, even if the interception values were slightly improved in this stage, they did not highlight increased concern for the defensive phase but were rather a result of the desire to score goals.

Our variables did not present collinearity. All values of the variance inflation factor (VIF) for each construct were less than 3. The F-Squared value (0.633) was higher than the accepted threshold (0.092), meaning that our model fit.

All the validation steps shown in Tables 3 and 4 and Figure 7 allowed us to consider that the indicators of the constructs were highly positively correlated, and the hypotheses H1 and H2 were accepted. Thus, we can assume that the circuit had a positive impact on the training and performance of football players.

**Table 4.** Multicollinearity.

| Statistics | Min F | Tackles F | Takeovers F | Interceptions F | Dribbles F | Assists F | Shots SF |
|---|---|---|---|---|---|---|---|
| VIF | 1.042 | | 1.466 | 2.478 | 1.402 | 2.256 | 2.721 |
| **Statistics** | **Min S** | **Tackles S** | **Takeovers S** | **Interceptions S** | **Dribbles S** | **Assists S** | **Shots SS** |
| VIF | 1.904 | 2.795 | 2.939 | 2.589 | 2.159 | 2.975 | 2.721 |

Circuit training has particular effects on the development of functional and morphological indices of motor qualities. Circuit training is not limited to a single method, but is an organizational methodological complex that includes several rigorously standardized exercise variants. However, its basis is the repetition of long series or interval exercises selected and combined into a well-organized structure [37,38].

From this perspective, the exercises used in a circuit can be selected from the category of means used in general physical training, gymnastics, or as basic preparatory exercises specific to sports branches. If the exercises that make up a circuit are varied (different motor skills), they are engaged in reciprocating all body segments without action, particularly one or other of the factors that differentiate the effort, thus achieving complex motor development and, therefore, multilateral physical training [22,23].

The age at which a circuit can be used in physical education lessons and sports training is between 10 and 12 years; the circuits consist of 5–6 exercises and gradually reach 8–10 exercises [39].

## 5. Discussion

The unimaginable situation in world sports in general and football in particular (given the specific conditions of competing, regardless of age) caused by the emergence of COVID-19 led all specialists in the field to design and apply a series of measures for all training components in order to maintain the highest levels of preparedness and play.

These coordinates were also the basis of this research, through which we tried to find a set of means customized and adapted to the circuit method with the purpose of preserving the sports form and avoiding a cumbersome, unspectacular style of play.

We will not reiterate the effects of applying the circuit method on sports training, but we must say that this method finds its applicability in several spheres of training (physical and technical) for both genders (considering their characteristics),even depending on the player position in the team.

The idea of implementing the circuit method is not new and is also applied to other levels, which is demonstrated by the variety of studies on this topic. The importance of the circuit method is given by its positive effects on improving technical–tactical aspects, but also physical components, which are so necessary in the game of football [40]. In this regard, Kumar (2020) assessed the effects of circuit training on the physical fitness qualities of speed, agility, and endurance based on the fact that players performed a sprint every 90 s during a football match [41].

Modern football provides more and more complex variants, reaching the point of creating circuits depending on player's position in a team. Apriliyanto et al. [42] (2017)

present training models based on the circuit method in order to improve the physical fitness of football goalkeepers.

In technical training, which is especially important for children and juniors, the circuit method has been successfully used over time, in the sense that players' technical levels are significantly improved. Matyas (2013) [43] conducted a study where technical circuits were used for young players aged 10–11 years.

In the literature on this topic, the use of the circuit method for children and juniors is limited to the development of strength and motor components in general, addressing less the improvement of technical and tactical aspects related to completing an action with a shot on the goal.

The results of our research showed that the investigated players managed to reach the final phase of shooting on the goal (due to their technical and tactical actions) more successfully in the second half of the championship rather than in the first one. If the means used in the circuit method are carefully planned and dosed and are designed in an attractive and stimulating way, they can contribute to the fulfillment of some stage objectives and, importantly, can lead to successful performance.

The weekly planning of the circuits proposed in the paper with dosages between 30 and 50 min, exceeded the content of basic technical–tactical assimilation, aiming to consolidate the techniques and tactical actions needed to play a modern football game at this age level.

The content of the exercises should be complex and as close as possible to the game model for the age group of 10–12 years, including driving a ball over different distances in a straight line, with a changes in direction or passing through obstacles, combined with passes, jumps, and turns before shooting on the goal. The difficulty of the circuits must comply with the methodological requirements, namely increasing step-by-step and allowing information transfer until the creation of automatisms.

Is it possible to design a specific training program that has an effect on technicaltactical indicators in the case of the post-pandemic period? We believe that this an be possible through implementing (designing) an optimal training model for children aged 10–12 years considering certain situations that the modern game demands.

An optimal training model developed in strict connection with a team's level, performance requirements, and training objectives can be generalized according to studies on a large number of teams and players, but it permanently imposes a certain order and content programming [44,45].

Can the number of means based on the circuit method be greater? We believe that it can, especially in situations similar to those in this study where the purpose for players was shooting on the goal and scoring goals, which require the ability to focus and to be creative and inspired.

Circuits with technical content close to that of our research can be developed in which shooting on the goal can be performed from crosses or combinations of 2–3 players (if they are in accordance with the objectives of the training process). Identifying performance differences between juniors at different stages can help develop prospective talent [46].

Consequently, the pandemic and the evolution of this state (from restrictions to changes during each pandemic wave) brought to the fore the formative necessity of creative and adaptive thinking. We cannot give up the work conducted over several years in the context of unforeseen situations without trying to find general and punctual solutions. Some athletes' careers are at stake, even if they are only in the training stage.

It should be noted that, after the study completion and the end of the championship with much improved results, which was also highlighted by the number of technical and tactical actions, a significant number of players (6) were transferred to top teams according to their age groups. This is another argument for the effectiveness of the circuit method applied to the game of football.

Professionals who work with children and juniors have a duty to test realistic and bold hypotheses in order to predict the progress of components with positive finality for achieving performance.

## 6. Conclusions

This innovative teaching and training method brought the team good results as quantified in motor skills, collaborative interactions, critical thinking, fast decision making, strategic overview, etc. Overall, in the 2021–2022 competition season, the team played 30 games (game duration: $2 \times 35$ min) and obtained very good results; thus, at the end of the first half of the championship, the team ranked eighth, won five matches out of fifteen, and had four draws and six defeats. In contrast, by the end of the championship, the team ranked third with nine wins, four draws, and only two defeats in fifteen matches.

We believe that using the circuit method suited the specific objectives of football training at this age. The exercises as part of this method complied with the age characteristics and were in accordance with the proposed type of effort. The dynamics of the exercises by their duration, intensity, and complexity, required as much from the players as the official game. This study is important for the continued training of physical education teachers and coaches (especially football coaches), who should develop the specific strengths of football players.

## 7. Limitations

This section is not mandatory but may be added if required by patents resulting from these indicators, even if our research did not focus on this direction through the degree of homogeneity of the group of athletes. We believe that the research results would be different if there were large differences between children in technical and tactical terms (this could be a possible study topic for the future).

This was a case study for the Progresul Bucharest team, which is why we did not assume the extrapolation of the results to other teams. We are considering a preliminary study conducted on several teams, and then, if the results are statistically demonstrated, they may be generalized to other groups of the same age.

**Author Contributions:** Conceptualization, D.G.P. and G.G.; methodology, D.G.P., L.I.P. and G.G.; software, D.G.P. and G.G.; validation, L.I.P. and G.G.; formal analysis, G.G. and D.G.P.; investigation, L.I.P.; resources, G.G.; data curation, D.G.P.; writing—original draft preparation, D.G.P., L.I.P. and G.G.; writing—review and editing, D.G.P., L.I.P. and G.G.; visualization, G.G.; supervision, D.G.P.; project administration, G.G. All authors have read and agreed to the published version of the manuscript.

**Funding:** This research received no external funding.

**Institutional Review Board Statement:** Ethical review and approval were waived for this study due to the fact that parents agreed to register and use the data.

**Informed Consent Statement:** Informed consent was obtained from all subjects involved in the study.

**Data Availability Statement:** Not applicable.

**Conflicts of Interest:** The authors declare no conflict of interest.

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
