# Peer review of "Circuit Football Training Customized for Young Players during and after the COVID Period"

_sustainability, doi:10.3390/su142416611_

Round 1

Reviewer 1 Report

Thank you for the opportunity to review this article by Dan Gheorghe Paun and colleagues, submitted to section: Sustainable Education and Approaches, Special Issue "IoT and Modern Technologies—Adequate Tools for Teaching and Practicing Physical Exercises in Time of Pandemic Crisis".

The topic is interesting and relevant to practicing physical activities during the COVID-19 period. The authors explain the steps of their approach based on the circuit method. The exercises used in the study are appropriated for 10-12 years girls, and for the specificity of the game of football at this level. The structure of the paper has a logical flow. The findings are properly addressed.

I only have some concerns/improvement suggestions related to the paper:

1.      I think it would be necessary that a short description of structure of the paper, by chapters, to be included at the end of the Introduction;

2.       The methodological approach is clearly described, but the images in figures 1-6 have not a good visibility. That is why, I would suggest that the authors increase their dimension or resolution;

3.       In order to improve the clarity of the paper, I would recommend:

(a). A description of the statistical hypothesis referred in line 235;

(b). A summary of the relationships among the 3 research variables moderated by the 2 proposed hypotheses;

(c) I believe that the hypotheses should be reformulated in order not to include the results in them;

(d) In my opinion, the research procedure should initially be clarified so that it can be verified and reproduced in other situations

(e). A questionnaire, specified in line 265, is used to “data collection … done by the coach / teacher during training and games/competitions” (lines 248-249). The paper specified three variables (Tour, Retour and Shots) with varying number of sub-items: Tour (8 sub-items); Retour (8 sub-items); Shots (4 sub-items). Existing validated questions would have to be adopted, where possible, in data collection process. Some sub-items could be self-developed to fit the paper context, taking into consideration existing literature. Please provide more details.

(f) A presentation of the statistical indicators obtained (average, standard deviation) would be welcome in order to be able to perform comparative analyzes on other samples as well;

4.      I have found some presentation / grammatical problems in the text:

-        The phrase  “Football has always been, is, and will always be the sport most spread worldwide” is used in lines 9 and 29;

-        Similar observation for the proposition “The consequences of these typological features are morphological, functional and psychological” from lines 72-73 and 83-84;

-        There are two phrases with the same meaning included in the lines 79-81 (“From a psychic point of view, at 10 - 12 years old, the player will focus on speed and skill exercises, avoiding long-term and boring efforts that require greater voluntary attention than a child can develop. “) and in the lines 84-86 (“Psychically, one shall focus on speed and skill exercises, avoiding the long-term and boring efforts requiring higher voluntary attention than a child may develop.”);

-        I think that “circus method” from line 138 would have to be replaced with “circuit method”;

-        The proposition from line 155: “The theoretical and methodical area that defines our field of activity approaches, from different angles, the issue of density in the training lesson” has no predicate;

-        Figures and tables would have to be included in the proximity of their references. In the paper, Table 2 is referred in line 241, page 6 and is included in page 8;

-        Table 2 is referred in line 241-242 before Table 1, referred in line 253;

-        Figure 8 is referred in line 301, but it is missing.

For these reasons, I would recommend the revision of the manuscript.

 I hope my feedback is useful to the author in improving the paper and wish him all the best in pursuing this important area of research.

Author Response

Thank you for all your recommendations.

Hereby we attach our answer.

Reviewer 2 Report

The introduction section is way too longer than required, should not exceed 2 pages. The introduction section should end with a study questions or research hypothesis. The main questions included in the Methods section should instead be listed as the last few lines of the introduction section. Teh study hypothesis mentioned in the Methods section should be moved to the last paragraph of the introduction section. The informed consent from participants and information on IRB approval needs to be included in the Methods section. 

The cut off for VIF of 5 seems too high, because the higher the value the greater the correlation with the other variable. Recommend redoing this correlation analysis using VIF value of 3 or less as the reference. This will lead to making more precise inferences from the data and will also result in a less inflated standard error. 

Review the rewrite the limitations section of the paper as it seems incomplete. The results are not generalizable to the entire population should be listed in this section and not in the results section. The discussion section is too small should be at least 2 pages long and require additional citations with comparisons to recently published literature. The manuscript requires a careful review and restructuring as many parts have information which is not needed and vice versa.  

Author Response

(The authors gave the same response as above.)

Round 2

Reviewer 1 Report

I would like to thank the authors for taking into consideration my comments and suggestions and revising their manuscript in order to improve the clarity, transparency and readability of their paper. The provided explanations clearly described how they have addressed each of my comments and what changes have been made.

I consider that the authors took into consideration all my concerns /recommendations.

I have only some minor review suggestions related to the:

-        Format of references 36, 37 and 40;

-        The missing year in references 37 and 40.

Once these are addressed, the paper can be accepted for publication.

Author Response

Thank you very much for all your support.

We made the requested changes.

Reviewer 2 Report

Format table 2 so that the column names (e.g., takeovers, position) are in one row similar to how they are presented in table 4.

Author Response

(The authors gave the same response as above.)
